# Top-KAST: Top-K Always Sparse Training

**Siddhant M. Jayakumar**
DeepMind
University College London

**Razvan Pascanu**
DeepMind
University College London

**Jack W. Rae**
DeepMind

**Simon Osindero**
DeepMind

**Erich Elsen**
DeepMind

## Abstract

Sparse neural networks are becoming increasingly important as the field seeks to improve the performance of existing models by scaling them up, while simultaneously trying to reduce power consumption and computational footprint. Unfortunately, most existing methods for inducing performant sparse models still entail the instantiation of dense parameters, or dense gradients in the backward-pass, during training. For very large models this requirement can be prohibitive. In this work we propose Top-KAST, a method that preserves constant sparsity throughout training (in both the forward and backward-passes). We demonstrate the efficacy of our approach by showing that it performs comparably to or better than previous works when training models on the established ImageNet benchmark, whilst fully maintaining sparsity. In addition to our ImageNet results, we also demonstrate our approach in the domain of language modeling where the current best performing architectures tend to have tens of billions of parameters and scaling up does not yet seem to have saturated performance. Sparse versions of these architectures can be run with significantly fewer resources, making them more widely accessible and applicable. Furthermore, in addition to being effective, our approach is straightforward and can easily be implemented in a wide range of existing machine learning frameworks with only a few additional lines of code. We therefore hope that our contribution will help enable the broader community to explore the potential held by massive models, without incurring massive computational cost.

## 1 Introduction

The Lottery Ticket Hypothesis [9] has spurred interest in training sparse neural networks [44], as it highlights a prior exciting result – that only a small subset of weights of a converged model are sufficient to represent the learnt function to high accuracy [14, 40, 29, 17, 36]. Perhaps even more exciting is the finding of Kalchbrenner et al. [17] that large sparse models outperform smaller dense models for a fixed parameter and floating point operation (FLOP) budget.

However, while encouraging, the primary method of finding such sparse subsets involves training a *dense* model. While there is a plethora of works proposing increasingly efficient ways to prune dense networks for sparse inference (dense-to-sparse training) [45, 27, 5], the field has only more recently begun to look at approaches that start training at the desired sparsity (sparse-to-sparse training) [26, 3, 28, 7].

Additionally, a high performance and scalable sparse-to-sparse approach would considerably benefit the democratisation of deep learning, as state-of-the-art models are ever increasing in size [34, 18, 39]. This increasingly leads to situations wherein state-of-the-art models require large clusters to train which most researchers would have limited access to. The large compute footprints and energy

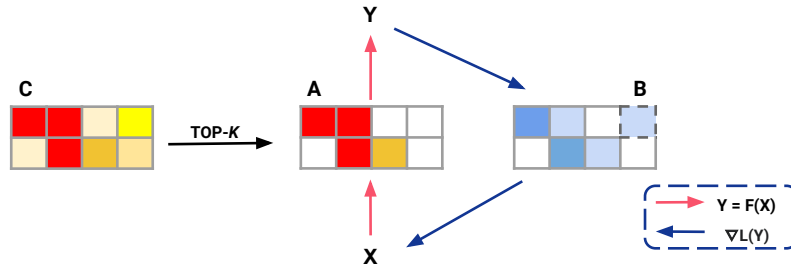

Figure 1: A diagramatic illustration of Top-KAST. While initialised with an effectively random mask, Top-KAST explores different permutations by updating an exploration set of weights and choosing the ones with greatest magnitude.

consumption of training such models also raises important environmental, moral and economic concerns [11, 33, 37].

State-of-the-art text-to-speech (TTS) [17, 1] and automatic speech recognition (ASR) [15, 31] are other domains that rely heavily on sparsity. Here sparse networks are used for efficient inference on embedded devices as well as to reduce latency. Further, enabling sparse-training could improve models' ability to personalize to different users, and maintain privacy on device [43, 23].

Sparse training requires both appropriate algorithms and software/hardware to take advantage of sparse operations. Whilst much of the focus in neural network training hardware has centred on accelerating dense linear algebra operations, there is already sparsity support in modern hardware [30] with more in the development pipeline [16].

Thus, a scalable and performant sparse-to-sparse method promises to unlock large potential benefits to neural network training — in terms of model scaling, reduced energy consumption and effective inference. The simplest and most scalable of these methods is to simply pick a random static sparse pattern at initialisation and train with this. Approaches such as Sparse Evolutionary Training (SET) [26] or Dynamic Reparameterization [28] improve on this by modifying their sparsity masks based on random evolution, but still lag behind corresponding dense-to-sparse methods. More recently, RigL [8] is able to match, or supersede the performance of dense-to-sparse methods. It does this by updating sparsity masks by using occasional gradient information. While theoretically entirely sparse, it is difficult to achieve RigL's theoretical bounds and avoid full dense materialization in common deep learning frameworks.

In this paper we aim to address some of these issues and propose a fully parameter-sparse training approach called **Top-KAST**. Our technique is scalable because it never requires doing a forward pass with dense parameters, nor calculating a dense gradient. It is also easy to implement within existing frameworks. Briefly, our method consists of selecting a subset of parameters $A \subset \Theta$ that correspond to the top-K parameters by parameter-magnitude for each training step, and applying gradients to a larger parameter subset $B \subset \Theta$ (where $B \supset A$.) To avoid the network fixating on a sub-optimal sparse subset, we introduce an auxiliary *exploration* loss to encourage the mask to adapt during training.

We find we are able to get state-of-the-art language modelling performance for small models, when training a Transformer-XL model using Top-KAST on the character-level task: enwik8 [24]. For image modelling, Top-KAST outperforms existing sparse-to-sparse training approaches, such as Sparse Evolutionary Training (SET) [26] and matches Rigging the Lottery (RigL) [7] on ImageNet across a range of floating-point operations (FLOPs) budgets.

## 2 Method: Top-KAST

The key desiderata for a **sparse training method**, is that it should:

1. Produce a network of desired weight sparsity $S_{final}$ after training is finished.
2. Have minimal compute and memory overheads relative to training a fixed (i.e. static) topology sparse model.

Dense-to-sparse training methods such as magnitude pruning, Dynamic Neural Wirings (DNW) [42] and Soft Weight Threshold Reparameterization (STR) [20] satisfy the first criterion but not the

second. Existing sparse to sparse methods satisfy the second constraint in different ways. SET and its derivatives occasionally prune unpromising connections and add new ones at random to maintain the same sparsity throughout training. RigL occasionally prunes unpromising connections and adds new ones based on the locations of the largest gradients from one mini-batch. We propose an alternate solution that still satisfies the second criterion and achieves high accuracy for a given number of training FLOPs while being easier to integrate into existing frameworks.

## 2.1 Sparse Forward Pass

We consider a generic neural network parameterised by function $f$ with parameters $\theta^t$ at some training step $t$ and input $x$. The output from the forward pass is $y = f(\theta^t, x)$. And during learning the parameters would be updated as $\theta^{t+1} = \theta^t - \eta \nabla_{\theta^t} L(y, x)$, where $L$ is the loss function.

Our aim is to to maintain a network weight sparsity of $S \in [0, 1]$ throughout training — where $S$ represents the proportion of weights that are zero ($D = 1 - S$ is the corresponding *density* proportion of the network). To do so, at each point in time we consider $\alpha^t$ – a parameterisation that retains a subset of weights from $\theta_i^t$, and replaces the rest with zeros. We have:

$$\alpha_i^t = \begin{cases} \theta_i^t & \texttt{if } i \in A^t \\ 0 & \texttt{otherwise} \end{cases}$$

with $A^t$ used to define a sparse subset of parameter indices that we consider to be "active" (i.e. non-zero) at time $t$. Membership of $A^t$ is restricted to the top $D$-proportion of weights (from $\theta^t$) by magnitude – that is:

$$A^t = \{i | \theta_i^t \in \texttt{TopK}(\theta^t, D)\}$$

In practice, we perform this top-$K$ operation *per layer* instead of on the flattened set of parameters[1]. One rationale for selecting weights according to their magnitude is that it is an effective but *inexpensive* estimate of which parameters contribute the most to defining the behaviour of the densely-parameterized function $f(\theta, x)$. Ideally we would like $f(\alpha, x)$ to be the best approximation of $f(\theta, x)$ using $\alpha$ of fixed sparsity-proportion $S$. To obtain insight into our approximation, we can examine the Taylor series expansion for $f(\alpha, x)$ around $\theta$, where $G$ is the gradient vector and $H$ is the Hessian matrix:

$$f(\alpha, x) \approx f(\theta, x) + G^T(\alpha - \theta) + \frac{1}{2}(\alpha - \theta)^T H(\alpha - \theta) + ...$$

While being able to calculate higher-order derivatives would provide more accurate sensitivity information [21], it is computationally intractable to do so for very large modern networks. However, as every term in the error scales with powers of $(\alpha - \theta)$, without any information about the higher order derivatives, minimizing the norm of $(\alpha - \theta)$ – which corresponds to our selection process – seems the best choice.

During learning we use $\alpha^t$ in both for the forward-pass and in the backward-pass – hence only incurring the inference and back-propagation compute costs of a sparse model. However, $\alpha^t$ is best thought of as a "temporary view" of the dense parameterisation, $\theta^t$. That is, the updates will be applied to $\theta$ rather than $\alpha$ and $\alpha^t$ will be reconstructed periodically from $\theta$ by the same deterministic procedure of picking largest (by magnitude) $D$-proportion of weights.

## 2.2 Sparse Backward Pass

The gradient of the loss with respect to a sparse $\alpha^t$ parameterisation need not result in a sparse gradient vector; indeed the gradient would typically be expected to be fully dense. This is because the gradients with respect to the 0 entries of $\alpha^t$ need not themselves be zero. This unfortunately would break our key desideratum (2). To avoid evaluating dense gradients we take inspiration from coordinate descent and compute the gradient for a coordinate block composed of parameters with indices from the set $B^t$, where:

$$B^t = \{i | \theta_i^t \in \texttt{TopK}(\theta^t, D + M)\}$$

By definition, $B$ is a superset of $A$ and contains the indices corresponding to the non-zero entries of $\alpha$ as well as an additional set of indices corresponding to the next largest $M$-proportion of entries (by magnitude) of the dense parameterisation, $\theta$. Updating the largest $(D + M)$-proportion of weights makes it more likely that this will lead to permutations in the top $D$-proportion weights that are active, and hence allows the learning process to more effectively explore different masks. We refer to this effective sparsity of $(1 - D - M)$ units as our *backward sparsity*.

Computing the gradient with respect to a subset of coordinates of $\theta$ implies that the gradient we are computing is sparse, and throughout the forward pass and backward pass we do not need to instantiate a dense vector of the size of $\theta$. The final update has the following form[2]:

$$\Delta_{\theta_i^t} = \begin{cases} -\eta \nabla_{\alpha^t} L(y, x, \alpha^t)_i & \text{if } i \in B \\ 0 & \text{otherwise} \end{cases}$$

At initialisation, $A$ will consist of a random subset of weight-indices from the freshly initialised $\theta^0$. As learning progresses, due to the updates on $B$ coming both from the primary loss and the auxiliary regularisation term (described in detail in the following section) this set will change and evolve the weights and topology most useful for the desired function approximation. We postulate learning as going through two stages (and this postulation seems to be observed in practice):

- In the first *exploratory stage*, at each iteration we select a different active set $A$, and its corresponding $\alpha$, and perform one update step on $\theta$ using gradients obtained from the loss on $f(\alpha, x)$ and the regularizer.
- In the second *refinement stage*, the active set $A$ effectively becomes fixed, as we settle on a stable pattern of non-zero weights which then undergo fine-tuning to their optimal values.

In the first stage, the updates on the "additional" coordinates in the set $B \setminus A$ allows exploration by changing the set of weights that will end up in the active set $A$ (and thus used in $\alpha$) on the next iteration. In the second stage, these "additional" updates will end up being increasingly less impactful and eventually will be effectively ignored, as they will not alter $A$ and hence will not be reflected in $\alpha$ for either the forward or backward passes. The *exploratory stage* of picking different subsets of parameters from $\theta$ sets makes our approach very different from simply having a fixed random sparsity pattern imposed on the model.

## 2.3 Exploration Regularisation Loss

The method outlined above may lead to a *rich-get-richer* phenomenon: with only the randomly selected weights at initialization being used if others receive insufficient weight updates for their norm to exceed the critical threshold. This problem may be particularly pronounced at high levels of sparsity, and to combat it we propose a heuristic inspired by the principle of *optimism in face of uncertainty*, widely used in reinforcement learning (RL) [4]. Concretely, we penalise the magnitude of the weights in set $B$, while those that are neither used nor currently being updated (set $C$) are not penalized at all. The net effect of this is to reduce the magnitude of the active weights, making it more likely that on the next iteration the algorithm considers new items for the membership of both set $A$ and $B$ — similar to how in RL, optimistic exploration adds bias to favour the selection of actions that have not thus far been chosen often.

We also posit that for high sparsity settings there is a teetering effect between weights in $B \setminus A$ and $A$ that are very close in magnitude, leading to a slow down in learning. We therefore propose to penalise $B \setminus A$ more than $A$ to increase the critical strength of updates needed for units from $B \setminus A$ to turn on and to stabilise the mask. We heuristically choose the scale to be inversely proportional to $D$, as this effect is more important for $D \ll 1$.

We express this penalty as an $L2$ regularisation, with a similar split of units as above[3]. Specifically:

$$Loss_R(\alpha_i^t) = \begin{cases} |\theta_i^t| & \text{if } i \in A^t \\ \frac{|\theta_i^t|}{D} & \text{if } i \in B^t \setminus A^t \\ 0 & else \end{cases}$$

### 2.4 Implementation of Top-KAST

As described above, the compute and memory requirements for Top-KAST in the forward and backward passes scale with the forward and backward sparsities, respectively. One possible concern is the additional cost of performing a `Top-K` operation in the forward pass every iteration. While the FLOPs required for this are much fewer than those needed by the actual training — this could necessitate fitting the dense model in memory. One way to alleviate this is to simply compute the the `Top-K` entries in parallel on CPU, thus avoiding the need to fit the model on the actual training hardware. The CPU could maintain the parameters in an appropriate data structure, such as a heap that would minimise the cost of updates. Lastly, we show in the sections below that the mask slowly stabilises and in fact we do not even need to perform this operation every step. In appendix C we show that we can get comparable results even if we perform this only *every* 100 *steps* which significantly reduces communication requirements and extra overheads.

## 3 Related Work

Methods that require dense weight or gradient information at training time but produce a sparse network at the end of training are now numerous and include: L0 regularization [5], variational dropout [27], discovering neural wirings [42], soft weight threshold reparameterization [20]. Magnitude Pruning is simple and effective [10] and we use it throughout as a baseline representative of this class of training methods. Such methods do not allow us to train larger sparse models than the biggest dense model we could train (in fact it is usually smaller due to overheads).

Sparse training of neural networks first happened through evolutionary means. Throughout the 1990s there was a flurry a research on the topic of Topology and Weight Evolving Artificial Neural Networks (TWEANNs) exemplified by [35]. While the networks *were* sparse during the evolution, this was not the focus of the research and the advantages of the sparseness in terms of enabling size and efficiency were mostly ignored. There has also been some recent work on using evolutionary methods to evolve sparse topologies [22].

Deep Rewiring [3] was the first work to consider sparse training of weight-sparse neural networks within the framework of gradient descent. It restricts weights to have a fixed sign, and sets weights to zero when their sign would flip. Additionally, it introduces a random walk in parameter space and can be thought of a constrained Monte Carlo sampling procedure over both the weights and the network connectivity. Despite theoretical convergence proofs, its practical performance seems to lag behind later, less well founded work [28].

This was followed by Sparse Evolutionary Training [26] which uses weight magnitudes to drop weights and introduces new connections at random, drawn from the original initialisation distribution. It is both simpler and more effective than Deep Rewiring. Our method, Top-KAST modifies the units based on gradient information instead which we find is more performant than random additions.

Dynamic Reparameterization [28] introduces a method for moving a parameter budget between different layers. This allows the network to better put parameter capacity where it is most effective. However, this ignores a FLOP constraint - the amount of FLOPs required to evaluate the network can change (usually upwards) because of these modifications.

Lastly, Rigging the Lottery (RigL) [7] is a recent and highly performant sparse-to-sparse method that matches or surpasses the performance of pruning-based methods. It uses infrequent full gradient calculations to decide which parameters to 'wake-up'. As it only requires knowing the location of the highest values of the gradients, its theoretical cost is proportional to the network sparsity, though this bound is hard to achieve in practice in current DL frameworks. We also compare Top-KAST to RigL in this paper and find we are able to perform comparably while alleviating the aforementioned implementation issues.

## 4 Experiments: ImageNet

Our aim in the section below is to demonstrate the efficacy of our method at enabling sparse training of models across different modalities (vision and language), model types (convolutions and attention) and different sparsity regimes. We start by demonstrating the efficacy of our method on the ImageNet dataset for image classification, where we train a sparse ResNet-50 as in previous works [7, 10]. This

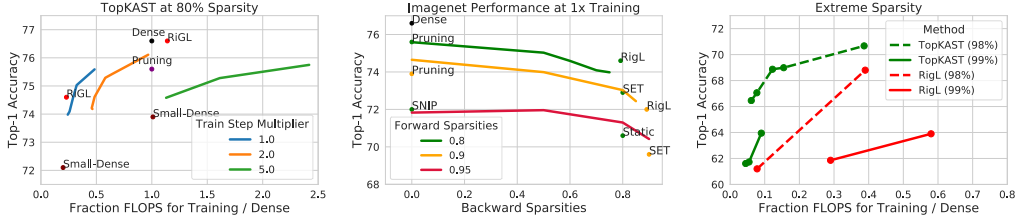

Figure 2: (a) FLOPS needed to train various sparse models as a fraction of those for a dense model. The FLOPS for Top-KAST vary as a function of the backward sparsity and the length of the training run. (b) Comparing methods on the basis of their backward sparsity. (c) Top-KAST and RigL compared at sparsities of 98% and 99%.

is a commonly used benchmark for sparsity methods, albeit often used in different regimes. We provide full details of model and hyper-parameters in the appendix B.

We first compare methods in the commonly used regime of fixed inference sparsity with first and last layers dense. As Top-KAST allows practitioners to choose their own level of backward and forward sparsity, we run Top-KAST for different levels of each, as well for multiples of the default training runs. We summarise this in Figure 2 above, showing the spectrum of performance versus FLOPS used (increases with decreasing backward sparsity and increasing training time), for a fixed forward sparsity of 80%. We also report results for a variety of standard and state-of-art methods.

We find (Figure 2 a and b) that Top-KAST is comparable (at constant FLOPS) to dense methods like pruning, while advantageously staying completely sparse throughout. Top-KAST also outperforms always-sparse methods like SET and Static random sparsity patterns. We further report results for sparsity levels 90% and 95% in 2(b) and results for relaxing the assumption of first and last layers dense, in appendix B.

**Comparing RigL and Top-KAST** Fig 2 also shows that the most performant prior sparse-to-sparse method is RigL and we see that Top-KAST performs comparably on a per-FLOP basis. RigL's update of its sparsity pattern requires *occasionally* calculating (a top-k over) dense gradients and in Fig 2 (b), we can see that when compared on the basis of average backward sparsity instead, Top-KAST requires slightly higher densities to match RigL's performance. However, while in theory RigL only needs the *highest* values of this dense gradient, it would require re-writing the gradient calculation for many primitives in existing DL frameworks to achieve this. Additionally, we note that RigL has many hyperparameters that might need tuning: when to start and finish updating the mask, how often to update, the initial drop fraction and the schedule by which this is annealed. On the other hand, Top-KAST requires no custom gradient calculations, and the only hyperparameter is the size of bucket $B$, and thus is easier to implement, to use, and is readily scalable. We expand on these implementation details in appendix section C. We also find in Fig 2 (c) that Top-KAST surpasses RigL at higher levels of sparsity (98% and 99%). Top-KAST's ability to choose slightly higher backward sparsities also means that at the cost of a little extra compute we are able to greatly increase performance.

### 4.1 Ablation studies

**Selection of $B \setminus A$.** We first consider the question of exploration in the backward pass and the method for selecting set $B$. We defined this set as those units used in the forward $A$ plus the next-highest set of units by magnitude. We can instead consider whether it would not be better to *randomly* sample these extra units. Intuitively we might explore more of the space and in expectation, allow gradient to pass through all units. We see in table 1 that this method is far better for sparsity of 90% but performs far worse for higher levels of sparsity, validating our choice. It is to be expected that this choice becomes more important in very sparse settings, where it would take many iterations to cover relevant weights if they are not directly targeted. Also, randomly picking additional weights means that the mask also changes more through training, whereas we expect the top-$k$ to stay more constant, thus reducing the potential cost of the sampling procedure.

**Analysing the learning dynamics** We can further test our hypothesis that our regularisation, combined with the learning dynamics, divides learning into an *exploration* phase, wherein an optimal

| Method | Sparsity Forward | Sparsity Backward | Top 1 Acc |
|---|---|---|---|
| Top-KAST | 0.9 | 0.8 | 73.03 |
| Top-KAST (Random) | 0.9 | 0.8 | **74.76** |
| Top-KAST | 0.95 | 0.9 | **70.42** |
| Top-KAST (Random) | 0.95 | 0.9 | 68.48 |
| Top-KAST ($t = 0$) | 0.9 | 0.0 | 68.26 |
| Top-KAST ($t = 5000$) | 0.9 | 0.0 | 72.05 |
| Top-KAST ($t = 16000$) | 0.9 | 0.0 | 74.14 |
| Top-KAST ($t = 32000$) | 0.9 | 0.0 | **74.65** |

Table 1: Ablation Experiments.

mask is discovered, and a *refinement* phase. To do so, we take a standard training run of 32000 steps and artificially stop the gradient updates to the 'extra' units not active in the forward pass ($B \setminus A$). We do so at different points of training (marked $t$ in Table 1) — start of training ($t = 0$), $t = 5000$, or halfway through. We find that removing all exploration units entirely ($t = 0$) is very harmful for performance, but training for just 5000 steps with these considerably boosts performance. At $t = 16000$ we have recovered most of the benefits of our method. This provides evidence that for the latter half of training, the gradients fine-tune performance on the learnt mask which stays more or less constant.

**Analysing the mask dynamics** We can further analyse how the mask changes through time. We take a standard training run as above with forward sparsity of $80\%$ and backward sparsity of $50\%$. We first measure the difference in the sparsity masks $m$ at pairs of points $5,000$ steps apart in training — i.e. $\frac{(m^t - m^{t+5000})^2}{|\theta|}$ — the fraction of units that change ($m = 1$ if the weight is active, else $m = 0$). This is summarised in figure 3 where we show the percentage change in masks across time (we plot min, mean and max across layers). We find that the mask indeed stabilises over time. We can further assess what units that are in set $C$ or the *reservoir* — units used in neither the forward nor backward passes at initialisation — ever turn on. We find that only about $5\%$ of these units are ever used and most of this change occurs at the start of training. This provides more evidence for the exploration and learning dynamics that motivate our design choices.

# 5 Experiments: Language Modeling

One class of models which have benefited hugely from a greater number of training parameters is language models, notably using the Transformer architecture [41, 32]. Language models predict the probability of strings of text, typically by tokenizing the text into a sequence of integers $x_0, \ldots, x_t$ (e.g. characters or words) and then decomposing the joint probability $p(x_0, \ldots, x_t)$ of this sequence into a product of conditional probabilities $p(x_0) \prod_{i=1}^{t} p(x_i | x_{<i})$.

Language model performance has been observed to follow a power-law of improvement when the data and model parameters are increased [18]. One challenge large parameter sets bring, is an increased strain on memory bandwidth to store the parameters. Approaches which can train and evaluate to

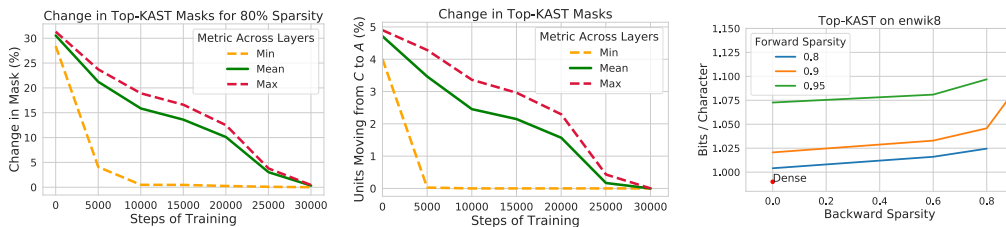

Figure 3: (a) shows that the mask gradually stabilises over time. (b) further, the number of units in set $C$ that make it to the active set $A$ is relatively small and also tends to 0.

comparable performance using less parameters can facilitate eventual training of larger models. We try Top-KAST to train language models on two commonly-benchmarked datasets: Enwik8 [24] which is a character-level benchmark derived from the *Hutter Prize* and WikiText-103 [25] which is a word-level language model benchmark. We use a long-range Transformer variant, the Transformer-XL [6]; training hyper-parameters are displayed in Supplementary Section A.

| Model | Params | BPC |
|---|---|---|
| Transformer-XL [6] | 277M | 0.99 |
| Stacked LSTM [12] | 21.3M | 1.67 |
| Hypernetworks [13] | 27M | 1.34 |
| mLSTM [19] | 46M | 1.24 |
| Transformer-XL [6] | 44M | 1.06 |
| All-Attention Transf. [38] | 39M | 1.01 |
| **Top-KAST** (80%, 0%) | **55M** | **1.00** |
| Top-KAST (80%, 80%) | 55M | 1.02 |
| Top-KAST (90%, 60%) | 27.7M | 1.03 |

Table 2: **Enwik8**: test BPC of small models.

| Fwd | Bwd | Params | Perplexity |
|---|---|---|---|
| 0% | 0% | 285M | 18.3 |
| 0% | 0% | 94M | 21.5 |
| 80% | 0% | 57M | 19.8 |
| 80% | 60% | 57M | 21.3 |
| 90% | 80% | 28.5M | 25.1 |
| 95% | 90% | 14.3M | 32.2 |

Table 3: **WikiText-103**: test perplexity for forward-backward sparsities.

On **Enwik8**, the baseline 24-layer dense Transformer-XL obtains 0.99 bits-per-character (BPC). We apply Top-KAST to training this model and vary the forward and backward sparsity rates as shown in Figure 3 (c). We find that we can obtain results comparable to the dense model all the way up to 80% sparsity. When comparing to previously published models that were trained and evaluated at a modest parameter count (under 60M parameters) in Table 2 we see that our Transformer-XL + Top-KAST achieves state-of-the-art performance. We also compare to magnitude pruning for a smaller Transformer model in appendix A.

On **WikiText-103** our baseline 16-layer Transformer-XL obtains 18.3 test perplexity. When trained with Top-KAST, we see in Table 3 that we can achieve 80% sparsity with minimal performance degradation, and performance begins to drift beyond the 90% sparsity range. Most importantly, the sparse model is significantly better than the even the smaller dense model with $3\times$ as many parameters.

# 6 Conclusion

In this work, we considered the question of effectively and efficiently training sparse neural networks. Performant sparse networks promise to democratise research with their low-resource usage, provide savings on compute and memory and also allow the proportional scaling up of model sizes. Prior works have shown the efficacy of pruning dense neural networks to highly sparse equivalents that are able to retain most of their original performance. Motivated by these successes, more recent works have attempted to maintain fully sparse networks throughout training. While a lot of progress has been made, most of these still involve the calculation of some dense weights or gradients, or involve operations that cannot be efficiently implemented with today's tools. Building on this, we introduced a novel method, Top-KAST that stays fully sparse in the both the backward and forward passes and is able to be implemented easily with modern neural network packages. Our method involves keeping around only the highest weights by magnitude in the forward pass and an extra set of *exploration* weights in the backward. Practitioners can choose their own values for both sparsities, based on the resource budget available. We further introduced a novel form of regularisation to encourage exploration in weight space. Coupled with this loss, Top-KAST achieves comparable performance to existing dense-to-sparse methods on ImageNet while remaining sparse, and exceeding the performance of several sparse-to-sparse methods. We further demonstrated the efficacy of our method on language modeling, the first such method to successfully sparsify Transformers in this context. We're also able to achieve state-of-art results for small models, with $1.00$ bpc at $55M$ parameters (versus a baseline of 0.99 at 277M parameters). While these are encouraging findings, more work is required to fully integrate Top-KAST with sparse hardware and the appropriate sparse kernels. We hope practioners and researchers alike find our method useful for reducing computational requirements, and to build on for even more powerful methods of sparsification.

## Acknowledgements

We'd like to thank Jacob Menick, Karen Simonyan, Tim Harley and Malcolm Reynolds for their helpful feedback throughout the project. We'd also like to thank Utku Evci for their help with running baselines for the ImageNet experiments.

## Broader Impact

Our work proposes a new method to train sparse neural networks that allows them to remain sparse throughout training – thereby enabling a practitioner to increase the model size that can be trained on a given piece of hardware. (This would also impact deployment too, in the case of on-device or real-time learning.) As we note in our introduction this scale-enabling should benefit the democratisation of deep learning since state-of-the-art models are ever increasing in size. Furthermore, there are beneficial impacts to be expected by reducing the computational footprint and energy consumption for training neural networks, as well as the higher-order impacts achieved if our work promotes the adoption of sparse networks more broadly – thereby also reducing the deployment/inference costs. While we do not expect any direct negative consequences from this work, the proposed method is general and widely applicable. We believe that the benefits offered by advances in machine learning net outweigh (by a significant margin) the potential risks and negative consequences. However, the technology as a whole is not purely good or benign. As one suggestion for future research building on our contribution, we would encourage colleagues who extend or apply our work to help us assess whether the inductive biases promoted by our sparsification methods have lead to any differential sensitivity to class imbalances or other aspects of the underlying data, relative to dense counterpart approaches for a given application. Since such issues could exacerbate problems related to algorithmic bias.

## Footnotes

[1]Either choice is valid and leads to the same number of parameters. Global pruning often increases the FLOP requirements by preferring parameters in earlier layers which have more reuse. It can also suffer from convergence issues at high sparsities due to differing scales in different layers leading to entire layers being pruned.

[2]Our approach is not a strictly valid coordinate descent method on either $\alpha$ or $\theta$.

[3]The gradient of the regularization term follows the same sparsity pattern as the gradient of the primary loss.

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
