[Supplementary Material]

# Supplementary

## A  Language Modeling

We train our Transformer-XL models with a very similar setup to Dai et al. [6]. The dense model hyper-parameters are listed in Table 4. We train with a learning rate warmup for 4000 steps from 1e-7 up to a value of 2e-4 and then apply a cosine decay. For WikiText-103 and enwik8 our dense model uses the same specification as the large Transformer-XL in Dai et al. [6], which has 285M parameters[4]

|  | Enwik8 | WikiText-103 |
|---|---|---|
| num layers | 24 | 18 |
| $d_{model}$ | 1024 | 1024 |
| $d_{ff}$ | 3072 | 4096 |
| $d_{embed}$ | 512 | adaptive: [2] |
| tie input/output embeddings | true | tie head only: [2] |
| num heads | 8 | 16 |
| dropout | 0.05-0.2 | 0.05 - 0.2 |
| learning rate | 2e-4 | 2e-4 |
| grad clip by global norm | 0.25 | 0.25 |
| window size | 768 | 512 |
| train mem size | 2304 | 768 |
| eval mem size | 5000 | 2000 |
| num params | 69M | 285M |

Table 4: Transformer-XL baseline hyper-parameters.

We further compare to a magnitude pruning baseline on enwik8. We found we were unable to implement this with the large model due to the additional memory requirements. Instead we compare Top-KAST and pruning on a smaller version of the Transformer-XL model of $69M$ parameters. This has identical training and hyper-parameters to below with the exception of $d_{model} = 512$, $d_{ff} = 1536$ and $num_heads = 8$. We summarise the results below. We find that pruning slightly outperforms Top-KAST when Top-KAST is allowed a dense backward (albeit the forward pass is also sparse). However, Top-KAST is competitive even in the regime of sparse backward passes.

| Fwd | Bwd | Params | Pruning BPC | Top-KAST BPC |
|---|---|---|---|---|
| 0% | 0% | 69M | 1.00 | 1.00 |
| 80% | 0% | 14M | 1.02 | 1.03 |
| 80% | 60% | 14M | - | 1.05 |
| 90% | 0% | 7M | 1.06 | 1.08 |
| 90% | 80% | 7M | - | 1.10 |
| 95% | 0% | 1.4M | 1.13 | 1.14 |
| 95% | 90% | 1.4M | - | 1.17 |

Table 5: **enwik8**: test perplexity for the smaller transformer model.

## B  ImageNet

For all ImageNet experiments we use a ResNet-50 set up as in prior work [10]. We use a batch size of 4096 and train for 32000 steps. We use use a learning rate of 1.6 (with a linear ramp up for 5 epochs) followed by learning rate drops by factors of 0.1 at 30, 70 and 90 epochs. For Top-KAST we use a weight decay of $1e-4$, and train for a range of backward and forward sparsity rates.

For our experiments we keep the first and last layers dense as in previous works [10, 7]. We also relax the assumption and show below the performance if all layers are sparsified.

All Layers Sparsified (1x Training)

Top-1 Accuracy

Forward Sparsities
— 0.8 (S)
— 0.9 (S)
-- 0.8 (D)
-- 0.9 (D)

Backward Sparsities

## C  Implementation of RigL and Top-KAST

In the sections above we compared briefly the implementations of RigL and Top-KAST and argued the relative ease of implementing Top-KAST because of some of the practical constraints a theoretically sparse implementation of RigL faces.

We first detail how RigL might actually be implemented and the difficulties that would be encountered. RigL occasionally requires calculating the `Top-K` values and locations of the full dense gradient with respect the parameters for every layer. The usual framework encapsulation is that all the gradients are computed and then sent to the optimiser. Doing the `Top-K` in the optimiser has the advantage of not needing modify the gradient calculations, but the large downside of meaning that the dense gradient would need to be materialised. This means the `Top-K` must happen inside the gradient calculation.

The type returned by the gradient calculation must be consistent, so it must always return both gradient values and locations and it must accept as arguments locations and a step count. If the step count indicates a `Top-K` over a dense gradient is to be performed, then input locations are ignored and the output locations contain updated locations. Otherwise, the input locations are used and simply copied to the output.

Inside the actual gradient calculation, it must 'chunk' the calculation of the dense gradient so as maintain a bound on the memory required. Assuming a data parallel regime, after each chunk is calculated locally, it must then be all-reduced. Then on each replica the running `Top-K` values are concatenated with the gradient chunk and a new running `Top-K` is calculated from this list. This process must proceed completely serially to maintain the memory bounds.

The serialisation introduces some perhaps non-trivial overheads, but most problematic is that no gradient calculations currently work like this. Every gradient calculation would need to re-written to do the appropriate chunking, this is both a high burden as this code involve rewriting a great deal of code. And it also introduces its own performance ramifications. Common libraries and/or data formats, especially for convolutions, might not support strides that would be necessary to compute arbitrary output shapes. If they do, it might come with negative performance implications.

Lastly, we show results for an implementation of Top-KAST that only requires calculating the `Top-K` every $N$ steps, where $N = 100$ (as opposed to $N = 1$, which corresponds to performing this every iteration). Such an implementation only requires occasional communication of the indices and weights and the `Top-K` operation can be calculated in parallel on CPU as it does not require any data or forward passes. The accelerator need only know the actual sparse weights and can be implemented entirely sparsely. We run Top-KAST for a variety of sparsity fractions and report the results below:

| **Fwd** | **Bwd** | $N = 1$ | $N = 100$ |
|---|---|---|---|
| 80% | 50% | 75.03 | 75.14 |
| 90% | 80% | 73.03 | 73.18 |
| 95% | 90% | 70.42 | 70.38 |

Table 6: Top-KAST at different frequencies of `Top-K`

## D  Pseudocode

In general Top-KAST can be implemented by modifying the parameters used in the forward pass and applying a gradient with respect to only some of the weighs in the backward pass. Below we demonstrate how this could be implemented with existing dense kernels and explicit masking of the weights. For a truly sparse implementation, custom sparse kernels would be required.

**Algorithm 1** TopKAST

```
// First perform a Top-K
dense_params = initialise()
fwd_params = TopK(dense_params, X%)
bwd_params = TopK(dense_params, Y%)
just_bwd_set = set(bwd_params) - set(fwd_params)
...
// Output with just the TopK params
output = model(fwd_params, input)
loss = loss_fn(output)

// Exploration L2 Loss
loss += l2(fwd_params) + l2(just_bwd_set) / (X/100)
...
// Update only the bwd params
bwd_params = bwd_params - grad(loss, bwd_params)
```

## Footnotes

[4]The original publication erroneously listed 255M parameters, however it has been clarified as 285M with the authors.