[Reviews · NeurIPS 2020]

Review 1

Summary and Contributions: This paper proposes a new method (Top-KAST) to train sparse neural networks, following the sparse-to-sparse training paradigm, which allows computational and memory benefits in both, training and inference. The main idea is to use two sparsity levels during training, a higher one for the feedforward pass and a smaller one for the backward pass. The latter one allows for using gradient information to search for a good sparse topology, while alleviating the computational requirements of previous work. The experimental evaluation shows good performance of the proposed method on various neural network models and in two main domains, computer vision and natural language processing.

Strengths: a) The proposed method is novel, simple, and efficient. b) It combines efficiently the magnitude based with the gradient based approaches for sparse training, taking the best parts from both approaches. c) The sparsity level can be fixed before training based on problem constraints. d) The proposed method solves the computational problem (of computing the gradients of all non-existing connections) of the state-of-the-art model in the sparse-to-sparse training paradigm (i.e. RigL - Ref 7) by looking just to a subset of the non-existing connections. e) Extensive experimental evaluation. f) On some of the language modelling tasks and in some conditions, the proposed method achieves state-of-the-art performance enabling sparse networks to outperform dense networks.

Weaknesses: a) Besides ImageNet, the paper would benefit for the computer vision domain, from experiments on other benchmark datasets (e.g. CIFAR 10/100). b) The design of the mask dynamics experiment does not consider the fact the hidden neurons are interchangeable in neural networks, and it seems to be suitable (even here up to some level) just for performing comparisons within the same model training (as it is practically done in the paper)

Correctness: The claims and the proposed methodology seem correct.

Clarity: The paper is well structured and very clear written.

Relation to Prior Work: In general, the relations and the differences to prior work are well discussed, with the exception of some minor comments which will be mentioned later.

Reproducibility: Yes

Additional Feedback: a) As the code is not provided (even if seems to be an easy implementation), I would suggest providing an algorithm/pseudocode in the paper to summarize somehow the proposed method, or alternatively to release the code. b) As a continuation of the mask dynamics experiments, as RigL uses all gradients information it would be interesting to see if the optimal sparse network discovered after training is the same at various runs with different random seeds. As Top-KAST uses just a subset of the gradients it would be interesting to see if it finds different optimal sparse networks at different runs, or if they are similar with the ones obtained by RigL. c) Consider adding experiments on CIFAR 10/100 at various sparsity levels. d) It seems that in a very sparse regime Top-KAST outperforms RigL in terms of performance (page 6, line 229). Which is the backward sparsity in this case, and do you have any hypothesis on why this is happening? Is it given by the Top-KAST randomized addition as yielded by the backward sparsity level? Perhaps, a systematic study to try understanding this phenomenon on a smaller dataset (e.g. CIFAR 10/100) would be helpful to perform during the rebuttal. Minor comments: e) page 5, lines 184 and 178. Please rephrase, as it is the other way around. According with arxiv.org, Sparse Evolutionary Training was released first, with 4 months before Deep Rewiring. f) As the precursor of RigL, this paper https://arxiv.org/abs/1907.04840 has to be discussed in the related work. g) Perform a proof-read of the paper. After rebuttal: I thank the authors for preparing the rebuttal. I am looking forward to the next version of the paper.


Review 2

Summary and Contributions: The authors present Top-KAST, a sparse-sparse training scheme that seeks to remove the overheads typically involved in generating a sparse network: training stages that remain dense. Instead of calculating dense gradients (RigL), or using a dense forward pass for some portion of training (many prior techniques), Top-KAST uses a sparse forwards pass and a less-sparse backwards pass. The extra parameters in the backwards pass (it is a superset of the parameters in the forwards pass) allow for the connectivity of the network to be learned, which typically happens early in training, before the fine-tuning the "necessary" weights in a second phase of training. The full set of parameters can be maintained in memory outside of the accelerator, and an occasional Top-K operation performed there before updating the sparse subset of those weights on the accelerator itself. Additional regularization terms (a) push the network to be more open to exploring with new weights not currently in the active subset and (b) keep the network from oscillating between sets of weights with similar magnitudes. Experiments are conducted on ResNet50/ImageNet and Transformer-XL with Enwik8 (BPC) and WikiText-103 (PPL). Starting with RN50: Comparisons with other techniques are shown for various FLOPS budgets and forward and backward sparsities. An ablation study shows the success of using a top-K metric for exploring other weights, as opposed to random sampling. A similar study shows that the extra parameters allowed in the backwards pass are helpful to the learning process, and indeed in two seemingly unique phases. Turning to Transformer-XL and the language modeling tasks, results are reported for a set of Top-KAST configurations in comparison to other small-model techniques on Enwik8, and similar results against the baseline and a small-dense version on WikiTex-103.

Strengths: The significance and novelty of this work, if the claims hold, is unquestionably high. Training a sparse model "from scratch" (or with constant sparsity, anyway) is of immediate use to many, researchers and practitioners alike. I'm not familiar with any work using quite this same technique, either, and it seems to be well-grounded in intuition. This work is very well-written and easy to read. The authors have done a fine job motivating the work and explaining the current state of research in this area. Similarly, the explanation of the Top-KAST technique itself was quite straightforward.

Weaknesses: As always, new hyperparameters (backward sparsity, training step multiplier [this seems to only be implied in Figure 2]) complicate things and make a new technique harder to adopt and use in practice. Are there any suggestions for how to choose the backward sparsity, given some target (forward) sparsity for the final network? The limited results could be much more convincing with the inclusion of more networks, as there's no direct theoretical motivation suggesting the Top-KAST approach should converge to a competitive model. While the concept is straightforward, as the authors have pointed out with RigL, the devil is in the details. I'd expect that building a system to actually train a network with a subset of parameters in the forward pass, a superset of that set in the backwards pass, and update and occasionally pull from a full complement of parameters in a remote memory might be more work than all but the most dedicated are willing to perform. Perhaps I'm wrong, and you can share the "few additional lines of code" required? If it were simple, I'd have expected to see a new SOTA result, no holds barred, for a giant Transformer. The FLOPS budget in Figure 2 seems to be theoretical, which is a good aspirational target, but turning that into runtime improvements isn't straightforward, especially given the efficient dense matrix operations most current accelerators provide.

Correctness: Line 217: "Fig 2 also shows that the most performant prior sparse-to-sparse method is RigL and we see that Top-KAST performs comparably on a per-FLOP basis." This doesn't seem to fall out from Figure 2, especially 2a, where RigL is nontrivially more accurate than Top-KAST for two different FLOPS budgets. Line 277: "We apply Top-KAST to training this model and vary the forward and backward sparsity rates as shown in Figure 3 (c). We find that we can obtain results comparable to the dense model all the way up to 80% sparsity." This isn't shown in the figure; can these results be added? Given the subjective nature of "comparable," skeptical readers will want to see the proof. Line 287: "Most importantly, the sparse model is significantly better than the even the smaller dense model with 3x as many parameters." I can't figure out how to make that true from Table 3. The rest of this section is similarly qualitative and subjective: "minimal performance degradation" is 1.5 PPL increase? "Performance begins to drift beyond the 90% sparsity range," indicating an increase in perplexity of 6.8 is not "drifting." One "ablation" or sensitivity study I'd like to see is what happens when changing the backward density. In the current ablation studies, you conclude that top-K selection is "far better [than random] for sparsity of 90% but performs far worse for higher levels of sparsity." This could be related not to the level of forward sparsity (which is what I assume you're talking about, given the mention of 90%), but instead to the level of backward sparsity. The results are shown for 0.9/0.8 and 0.95/0.9, but what about 0.95/0.8?

Clarity: Generally, yes - only a small handful of typos/grammar/formatting issues: 258: across layer -> across layers 270: one challenge … , is an increased --> one challenge … is an increased 452: LaTeX formatting issue on num_heads Various (217, 228, 237): capitalization should be consistent for section/table/figure references, and abbreviations (or lack thereof) for Fig/Figure.

Relation to Prior Work: Yes, excellent job.

Reproducibility: Yes

Additional Feedback: Does Top-KAST decay into any other technique if backward sparsity == [forward sparsity, 0]? These seem to be corner cases you use several times, I'm curious if there's anything interesting to say about them. I'm curious what happens for more moderate sparsities for not-giant networks, for which pruning matches dense accuracy - can Top-KAST match the accuracy in this regime, which would mean there's no reason to ever not train with Top-KAST? (Would it actually mean that in practice, or are there considerations that would make standard dense training still preferable?) In general, this is a very important area and an interesting technique that holds promise. I'd really like for some of the qualitative/subjective language to be tidied up to avoid drawing conclusions that may not be supported by the data, though. While I could see how to reproduce the major results of this work, the major results don't seem to include "training a model larger than will fit in accelerator memory," as evidenced by all the fully dense baselines. So, while that's a good thing for the reproducibility metric, it calls into question the validity of some claims. ---------- UPDATE ---------- I appreciate the authors' response to my review: addressing backward sparsity selection, agreeing to review/temper some overzealous claims, and comparing random selections vs. top-k. These weren't my main criticisms, however. Seeing pseudocode to understand the practical simplicity of the technique (as claimed), and broadening the empirical results (it'd be okay to put as many details as possible in the appendix, as long as the main results show up in the first eight pages; I understand space limits) are still keeping me from upgrading my rating. Additionally, at least one other reviewer pointed out that the main text didn't have any direct comparison to dense-to-sparse accuracy. While it's not apples to apples, and the two approaches have slightly different goals, having these numbers in the main text would provide useful context.


Review 3

Summary and Contributions: This paper proposes Top-KAST a sparse-to-sparse training algorithm for DNNs, such that during training time there is only a small part of network trained stochastically and the final model is sparsified. The experiments are performed on language modeling and image classification tasks.

Strengths: The sparse to sparse network pruning is an interesting problem. The proposed approach makes intuitive sense.

Weaknesses: My main concerns of the paper are: (1) limited novelty, (2) insufficient evaluations. Please see my detailed comments below.

Correctness: Partially and should be improved.

Clarity: yes

Relation to Prior Work: yes

Reproducibility: No

Additional Feedback: First of all, the proposed method is intuitive. However, it lacks theoretical justification, and tuning the hyperparameters such as B, M% is quite empirical (without convergence guarantees). Also, the method is a simple tweak of weight magnitude based pruning and gradient-based weight expansion of previous methods. Therefore, the novelty is limited. Evaluation: 1. The paper compared Top-KAST with other sparse-to-sparse pruning algorithms. However, there is no comparison with dense-to-sparse algorithms. Particularly, there are many SOTA dense-to-sparse pruning algorithms for ResNet50 on ImageNet dataset with impressive sparsity and close-to original accuracies. The paper should include the SOTA dense-to-sparse algorithms to provide a solid performance (sparsity and accuracy) reference. 2. On WikiText-103 dataset, there are significant accuracy losses with Top-KAST. Are there any explanations? Also, there is no dense-to-sparse baseline provided. Therefore, it’s unclear how competitive the proposed method is. 3. The paper lacks sufficient details on how the experiments are performed even after reading through the appendix. So reproducibility is an issue too.


Review 4

Summary and Contributions: After author response: Thank you for clarifying about the baselines, I didn't notice that these baselines were only embedding in Figure 2, rather than in the tabular results. I'd encourage the authors to make this comparison a bit more explicit in the text, as well as the reasoning for excluding some baselines. I've updated my scores accordingly. === The authors propose an approach for sparse NN training which applies top-k sparsity to model parameters in both the forward and backward pass. They evaluate this approach on ImageNet classification and language modeling and find that they are able to achieve high levels of sparsity with minimal drop in task accuracy.

Strengths: The approach is simple and would be reasonably easy to implement. It achieves only slightly worse performance than some densely trained baselines.

Weaknesses: - Poor comparison to related work. No quantitative comparison to several sparse-to-sparse training methods (e.g., SET, DSR). There is very limited comparison to dense-to-sparse methods, namely magnitude pruning, although this is relegated to the Appendix and shows that Top-KAST underperforms magnitude pruning. - Limited practical value. Efficiency claims are based on theoretical FLOP reduction, but there is no discussion or empirical evaluation of efficiency gains in practice. Based on my understanding, this approach seems to be slower to train than a dense model (due to the top-k and inefficiency of unstructure sparsity on modern training hardware), and achieves worse task accuracy than dense-to-sparse approaches (e.g., magnitude pruning). There may be memory efficiency gains, but these are not clearly evaluated (e.g., how is optimizer state handled for Adam when training Transformer XL?). - The methodology is not properly evaluated or justified. For example, the proposed "exploration regularisation loss" term is introduced in Section 2, but then never mentioned again. What is the impact of that term? There is also no theoretical justification provided for why training with top-k sparsification of weights and gradients works.

Correctness: The poor evaluation makes it difficult to fully trust the claims. Additionally, the empirical results would be stronger if the authors did repeated runs and reported the variance across seeds.

Clarity: The writing is clear and the paper is well organized.

Relation to Prior Work: No, see comments under Weaknesses and additional comments below.

Reproducibility: No

Additional Feedback: - I don't fully follow the explanation on lines 221-223 for why RigL requires rewriting the gradient calculation. It seems like the gradients can be materialized on a per-layer basis and then discarded. It should be possible to do this in PyTorch with a custom autograd Function wrapper, similar to gradient checkpointing (torch.utils.checkpoint.checkpoint). - In Table 1, what is the variance of the results across runs if you change the random seed? I also have same question for the results presented in Figure 3. - In the third section of Table 1, why is Sparsity Backward set to 0.0? Wouldn't it be more consistent to do this ablation with a backward sparsity of 0.8? - In Table 2/3, please include more settings where forward sparsity and backward sparsity are equal (e.g., Fwd = Bwd = 80%). This would help illustrate any gains you achieve by decoupling these values. - In Figure 2(a), what is "train step multiplier?" Does that mean you're training for twice as many steps? Perhaps reword to "# train steps" and have "1x", "2x", "5x" - The rightmost pane of Figure 3 is not described anywhere. - What is the value that's reported in Table 6? Is it Top-1 accuracy? - How does your approach compare to meProp [1] or "Sparse Weight Activation Training" (arxiv.org/abs/2001.01969)? - typo (line 170): "do not allow us train" -> "do not allow us to train" [1] Sun, Xu, et al. "meProp: Sparsified back propagation for accelerated deep learning with reduced overfitting." ICML 2017.

[Author Response · NeurIPS 2020]

We thank the reviewers for the time and effort they put into the reviews and address their questions and comments
below. In particular R3 and R4 asked for more information about baselines and we point them below to the relevant
baselines in the paper (we compare to 6 different methods) and try our best to address their concerns about the method.

**re CIFAR/other architectures and code (R1/R2)**: We agree that more datasets/architectures would always be useful
however in practice CIFAR is easier to sparsify and train sparse than ImageNet, thus we believe that ImageNet is
a better demonstration of the capabilities of our algorithm. We would argue that CIFAR should not be used for
comparisons of these techniques as findings do not always translate to larger datasets. We provide results from two
completely different/popular families of models (resnet and transformer) and two different tasks (language modelling
and classification) on large scale datasets and believe this demonstrates the efficacy of the method. We will add
pseudocode for how to implement this method. In general this can be implemented with a custom getter in TensorFlow,
such that the weight matrix is sparsified before any computation occurs.

**re tuning backwards sparsity (R2)**: In practice the correlation between backward sparsity and performance is
straightforward: lower backwards sparsity generally allows us to discover better sparsity masks. A benefit of this aspect
of our method's design is that backwards sparsity can thus be chosen based on the FLOP budget available and the amount
of memory available – in general the recommendation would be to use as dense a backward pass as the budget allows.

**Re baselines (R3/R4)**: Both reviewers ask for a comparison to dense-to-sparse methods. We point the reviewers
to Figure 2 in the paper **where we compare with six different baselines, of which two (Pruning and SNIP) are**
**dense-to-sparse methods**. Moreover we outperform most of them on a per FLOP basis while staying entirely
sparse. Reviewer 4 asks for comparisons to SET and DSR. We compare to RigL which consistently outperforms
both SET and DSR so we did not see the value in including this baseline, although we do also compare to SET
in Figure 2 and are happy to add DSR.

**Re Novelty and Tuning (R3)**: We disagree that the model is a simple tweak of pruning given we maintain constant
sparsity throughout training. Moreover we match state-of-the-art performance on Imagenet and sparsify TransformerXLs
– something which has thus far not been done in a sparse-to-sparse manner. On the point about tuning hyperparameters:
we in fact reduce the hyperparamtere compared to pruning (which requires a hand-tuned schedule). Additionally our
results outperform or are similar to those published by other sparse-to-sparse methods and are also the first method
that allows differing backward and forward sparsities.

**Re Wiki103 (R3)**: There is no published method that has been applied to Wiki103 in a completely sparse-to-sparse
manner. We also disagree that there are huge performance drops - as we show **we match the performance of a 97M**
**parameter dense model with 57M and 60% backward sparsity**. On the point about comparing to pruning, we
do this on a smaller model (as we could not fit the pruning masks for bigger models in memory) and compare
in Table 5 in the appendix.

**Re theoretical flops (R4)**: Every single published method we compare to also relies on theoretical FLOP reduction.
Current industry trends strongly suggest that future advancements in hardware/software kernels will allow us to
implement sparse methods with native-sparse support. While valid, the above criticism can be equally levied against
every previously published sparse-to-sparse method and would devalue every single published paper in the field.

**re theoretical proofs/limited practical value(R4)**: We do not have a proof that using top-k is a best/better choice
and is unclear whether this is tractable theoretically. We do however point the reviewer to section 2.1 where we
explain with a taylor approximation why this approximation holds and that the algorithm will converge to some local
minima. On structured sparsity: our method is equally applicable to block sparsity given a function such as max-or
sum-of-absolute-values that reduces a block to a scalar value.

**re regularization loss (R4)**: this is used in all experiments and is an integral part of Top-KAST (not a potential
add-on). We will make this clearer in the paper.

**re correctness and other experiments (R2)**: We thank the reviewer for their comments and suggestions – on random
vs topk, we will explore this further. One added benefit of top-k over random is also the stability of the mask and not
having to constantly resample it. On corner cases, bwd sparsity=fwd would be similar to SET with a difference in how
we sample new parameters and bwd=0 would be like DNW but with the special regularization. On the comparison
with RigL and some of the subjective words, we will further clarify our language in the paper as our intention was
not to overclaim. In general for lower values of sparsity, RigL outperforms Top-KAST, and Top-KAST outperforms
for higher. We view the overall results as comparable (specifically both methods significantly outperform previous
baselines like SET) and note in the paper the main advantages of Top-KAST over RigL in section 4.

**Typos and minor comments and ablations**: We thank all reviewers for their notes on these and will correct this
in future versions. We thank R1,R2 and R4 for their suggestions for ablations. While unfortunately out of scope
for this document, we will strive to include this in the paper.

[Meta-Review · NeurIPS 2020]

After much discussion, the reviewers largely converged towards recommending to accept this submission. The reviewers were satisfied with the authors' response, and have updated their reviews accordingly. In my decision I reduced the weight I gave to the score of R3 as they did not justify whether the author response addressed their criticism or not.